# Glycerol-Induced Powdery Mildew Resistance in Wheat by Regulating Plant Fatty Acid Metabolism, Plant Hormones Cross-Talk, and Pathogenesis-Related Genes

**DOI:** 10.3390/ijms21020673

**Published:** 2020-01-20

**Authors:** Yinghui Li, Lina Qiu, Xinye Liu, Qiang Zhang, Xiangxi Zhuansun, Tzion Fahima, Tamar Krugman, Qixin Sun, Chaojie Xie

**Affiliations:** 1Key Laboratory of Crop Heterosis and Utilization (MOE) and State Key Laboratory for Agrobiotechnology, Beijing Key Laboratory of Crop Genetic Improvement, China Agricultural University, Beijing 100193, Chinaqxsun@cau.edu.cn (Q.S.); 2Institute of Evolution, University of Haifa, 199 Abba-Hushi Avenue, Mt. Carmel, Haifa 3498838, Israel; 3Ministry of Education Key Laboratory of Molecular and Cellular Biology, College of Life Sciences, Hebei Normal University, Shijiazhuang 050024, China

**Keywords:** disease resistance, glycerol application, *Triticum aestivum* L., transcriptome sequencing

## Abstract

Our previous study indicated that glycerol application induced resistance to powdery mildew (*Bgt*) in wheat by regulating two important signal molecules, glycerol-3-phosphate (G3P) and oleic acid (OA18:1). Transcriptome analysis of wheat leaves treated by glycerol and inoculated with *Bgt* was performed to identify the activated immune response pathways. We identified a set of differentially expressed transcripts (e.g., *TaGLI1*, *TaACT1*, and *TaSSI2*) involved in glycerol and fatty acid metabolism that were upregulated in response to *Bgt* infection and might contribute to G3P and OA18:1 accumulation. Gene Ontology (GO) enrichment analysis revealed GO terms induced by glycerol, such as response to jasmonic acid (JA), defense response to bacterium, lipid oxidation, and growth. In addition, glycerol application induced genes (e.g., *LOX*, *AOS*, and *OPRs*) involved in the metabolism pathway of linolenic and alpha-linolenic acid, which are precursor molecules of JA biosynthesis. Glycerol induced JA and salicylic acid (SA) levels, while glycerol reduced the auxin (IAA) level in wheat. Glycerol treatment also induced pathogenesis related (*PR*) genes, including *PR-1*, *PR-3*, *PR-10*, *callose synthase*, *PRMS*, *RPM1*, *peroxidase*, *HSP70*, *HSP90*, etc. These results indicate that glycerol treatment regulates fatty acid metabolism and hormones cross-talk and induces the expression of *PR* genes that together contribute to *Bgt* resistance in wheat.

## 1. Introduction

Bread wheat (*Triticum aestivum* L.) is one of the most important food crops worldwide. *Blumeria graminis* f.sp. *tritici* (*Bgt*), is an obligate biotrophic ascomycete fungus that invades the aerial parts of wheat, causing powdery mildew disease. The damage caused by *Bgt* can result in yield losses from 30% to 40% in years with severe epidemics [1,2]. Deployment of resistance (*R*) genes and spraying with foliar fungicides are the common methods for diseases control in wheat. However, many *R* genes lost their effectiveness due to the appearance of new pathogen races, and fungicides are not always economically feasible and environmentally friendly. Therefore, new sustainable effective ways for disease control in wheat are needed.

Plants have been developing multifaceted innate immunity systems along the co-evolution of plants and pathogens. When plants detect an attempted pathogen invasion, the transmembrane pattern recognition receptors (PRRs) recognize pathogen-associated molecular patterns (PAMPs), resulting in PAMP-triggered immunity (PTI) [3]. The other immune response acts largely within the cell, using NOD-like receptors (NLRs) that specifically recognize fungal effectors, resulting in effector-triggered immunity (ETI) [3,4]. After recognition, several downstream signaling events are elicited, including reactive oxygen species (ROS) accumulation, and transient activation of mitogen-activated protein kinases (MAPK) signaling cascades, as well as interaction with some plant hormones including salicylic acid (SA), jasmonic acid (JA), auxin (IAA), abscisic acid (ABA) and other phytohormones [5,6,7].

Glycerol-3-phosphate (G3P) and oleic acid (OA18:1) are two important signal molecules associated with resistance to fungal pathogens in plants [8,9,10]. *Arabidopsis* plants overexpressing *GLI1*/*NHO1* (encoding a glycerol kinase) increase G3P levels and enhance resistance against bacterial disease caused by *Pseudomonas syringae* [11]. *Arabidopsis* mutation *gli1* reduces the G3P level and enhances susceptibility to *Colletotrichum higginsianum* [12,13]. It was suggested that G3P may contribute to plant resistance. However, *ssi2* (encoding a stearoylacyl carrier protein fatty acid desaturase) mutants with lower OA18:1 levels elevate SA and JA levels, induce the expression of pathogenesis-related (PR) proteins, and improve plant resistance [8,14,15,16]. Several reports show that lipids are a major source of organic carbon delivered to the fungus [17,18]. Mutualistic mycorrhizal fungi recruit the plant fatty acid biosynthesis program to facilitate host invasion; with the case of *kas1*, *kar1*, and *fatb-1* mutant plants of fatty acid biosynthesis enhance disease resistance compared with wild type plants [19]. Therefore, these lipids, as plant signals or fungal organic carbon sources, play an important role in the host–pathogen interactions of plants and pathogenic fungi.

Exogenous glycerol increases G3P level, with a reduction in OA18:1 by acylation of G3P with OA18:1, resulting in the induction of *PR* gene expression and conferring resistance to different pathogens in *Arabidopsis*, rice, soybean, and cacao tree [15,20,21,22]. Our previous studies indicated that glycerol can induce resistance of wheat to *Bgt* with increased G3P and reduced OA18:1 levels, and accumulation of SA, JA, and ROS levels [23]. Moreover, glycerol appears to have some potential to be applied in wheat fields as an environmentally-friendly agricultural chemical to help manage *Bgt* diseases [22,23]. However, the glycerol-mediated disease resistance pathways have not yet been clarified.

Transcriptome and proteomics technologies provide powerful tools for identification of genes and pathways associated with plant disease resistance. Earlier studies using Affymetrix wheat array reveal 3014 and 2800 wheat genes that are involved in the interaction of *Bgt* with susceptible cultivar Jingdong 8 and the resistant near-isogenic line carrying *Pm30*, respectively. The genes’ involvement in lipid degradation and metabolic processes, fatty acid catabolic process, and plant hormone signal transduction pathways have been identified [24]. Using iTRAQ-based quantitative proteomics, 394 proteins were identified in response to *Bgt* in a resistant wheat line, including *PR* polypeptides, oxidative stress responsive proteins, and primary metabolism pathways [25]. However, these proteins represent less than 10% of the differentially expressed genes that are identified at different time points post *Bgt* infection, detected by RNA sequencing [26].

In the current study, wheat plants treated with glycerol were resistant to *Bgt*, while water-treated plants were susceptible. Differentially expressed genes (DEGs) were identified by using RNAseq analysis. We found that glycerolipid metabolism and fatty acid biosynthesis pathways were in response to *Bgt* infection, which might contribute to G3P and OA18:1 accumulation. In addition, glycerol regulated hormone cross-talk (increased JA and SA levels, reduced IAA level) and induced *PR* proteins in wheat, are probably contributing to resistance to *Bgt*. This study contributes to a better understanding of the glycerol-mediated disease resistance pathways in wheat.

## 2. Results

### 2.1. Bgt Resistance Test in Water-Treated and Glycerol-Treated Wheat Leaves

Four groups of wheat plants were prepared for the identification of the transcriptional responses to glycerol treatment of wheat leaves, with and without *Bgt* infection: H0 and G0 groups represented water or glycerol-treated plants without *Bgt* infection; H24 and G24 groups represented water or glycerol-treated plants with *Bgt* infection (Figure 1A). The morphology of conidiophores and hyphae was monitored in water and glycerol-treated wheat leaves infected with *Bgt* (H24 and G24 groups). The obtained results showed that ~90% of the spores successfully germinated on water-treated leaves, while only about 10.7% of the spores germinated on the glycerol-treated leaves (Figure 1B,C). These results suggest that glycerol treatment induced effective resistance against *Bgt* infection. Therefore, changes in gene expression induced by both glycerol treatment and *Bgt* infection were explored using whole transcriptome analysis.

### 2.2. Transcriptome Sequencing, Assembly, and Functional Annotation

Transcriptome sequencing revealed a total of 681,705,618 clean reads (85.21 G clean data) from twelve cDNA libraries, constituting 41,214,380–77,987,478 clean reads (5.15–9.74 G clean data) for each pool, with a quality score of Q30 ≥89.12%. Following assembly, 81.99%~86.69% of the reads were mapped on the wheat genome reference sequence IWGSC RefSeq v1.0 (Appendix A). In total, we annotated 106,755 distinct assembled unigenes after blast searching in six databases (Appendix A, Figure 1D).

### 2.3. Identification of Differentially Expressed Genes (DEGs)

In the water-treated group (H24 vs. H0), 5616 DEGs were identified of which 3489 were upregulated and 2127 were downregulated in response to *Bgt*. In the glycerol-treated group (G24 vs. G0), only 2494 DEGs were identified, of which 1723 were upregulated and 771 were downregulated in response to *Bgt*. A total of 414 DEGs (376 up, 38 down) were identified in the G0 vs. H0 groups, and 463 DEGs (308 up; 155 down) were identified in the H24 vs. G24 groups (Table 1). A total of 1317 upregulated and 467 downregulated DEGs were overlapped in H24 vs. H0 and G24 vs. G0 groups, respectively. More than half of the DEGs in G0 vs. H0 groups overlapped with G24 vs. G0 or H24 vs. H0 groups (Figure 1D). DEGs identified in different biological replicates were clustered together in a heat map of expression levels, indicating good reproducibility between replicates (Appendix A). The annotated DEGs of different groups (H24 vs. H0, G24 vs. G0, G0 vs. H0, G24 vs. H24) are attached in Appendix A.

### 2.4. Expression Patterns in Response to Bgt Infection (H24 vs. H0 and G24 vs. G0)

To better understand the response to *Bgt* infection in both water- and glycerol-treated plants, we annotated DEGs into Gene Ontology (GO) terms and KEGG pathways (Appendix A). DEGs were mainly classified into cell killing, immune system process, response to stimulus, signaling, and metabolic process GOs (e.g., biological process terms, Appendix A). Those upregulated DEGs both in H24 vs. H0 and G24 vs. G0 groups showed significant enrichment in plant–pathogen interaction, sphingolipid metabolism, and phenylalanine metabolism pathways (Figure 2A,B); while the linoleic acid metabolism pathway was only markedly enriched in upregulated DEGs of H24 vs. H0 groups (Figure 2A). However, those downregulated DEGs both in H24 vs. H0 and G24 vs. G0 groups showed significant enrichment in photosynthesis-antenna proteins, steroid biosynthesis, fatty acid elongation, starch, and sucrose metabolism, and sesquiterpenoid and triterpenoid biosynthesis showed significant enrichment (Figure 2C,D). KEGG-enriched analysis showed that 36 DEGs in H24 vs. H0 groups and 22 DEGs in G24 vs. G0 groups belong to plant hormone signal transduction pathways. Those DEGs are involved in the signaling pathways of auxin (*AUX/IAA*), cytokinin (*AHP* and *A-ARR*), abscisic acid (*PP2C* and *SnPK2*), ethylene (*ETR*, *EIN2*, *EIN3*, and *ERF1/2*), and salicylic acid (*NPR1*, *PR-1*) (Appendix A). Overall, 66 DEGs in H24 vs. H0 groups and 40 DEGs in G24 vs. G0 groups were identified in the plant–fungus interaction pathway. Among them, *PPRs* (CEBiP), *FLS2*, *MKK1/2*, *WRKY33*, *pti1*, *NHO1*, and *PR1* were induced by *Bgt* in the H24 vs. H0 groups, and *CaMCML*, *MKK1/2*, *WRKY33*, *NHO1*, and *PR1* were induced by *Bgt* in the G24 vs. H0 groups, which might be associated with the PAMP-triggered immunity. Moreover, *RPS2* and *HSP90* were induced by *Bgt* in H24 vs. H0 groups, and *RPM1*, *RPS,2* and *HSP90* genes were induced in response to *Bgt* in G24 vs. G0 groups, which may be involved in effector-triggered immunity (Appendix A).

### 2.5. Changes in Expression Patterns of Glycerolipid Metabolism and Fatty Acid Biosynthesis Pathways in Response to Bgt Infection

We analyzed the DEGs of glycerolipid metabolism and fatty acid biosynthesis pathways, which responded to *Bgt* infection in H24 vs. H0. We found that 13 genes from 265 genes which were involved in glycerolipid metabolism pathways were regulated by *Bgt* infection (Figure 3, Appendix A). Among them, three *TaGLI1* homologous genes (2.7.1.30), identified as glycerol kinase from A, B, and D genomes of wheat, were upregulated by *Bgt* infection. In addition, 1.1.1.21 identified as an aldehyde reductase, which transfers D-glycer-aldehyde to glycerol, was also upregulated by *Bgt* infection. While, 2.3.1.15 (named as *TaACT1*) identified as glycerol-3-phosphate acyltransferase was downregulated by *Bgt* infection. Moreover, 11 DEGs from 265 genes which were involved in fatty acid biosynthesis pathways were regulated by *Bgt* infection (Figure 3, Appendix A). We found that three *TaSSI2* genes (1.14.192) from A, B, and D wheat genome, and some upstream genes identified as *FabI*, *FabZ*, *FabF*, and *FabD* were induced by *Bgt* infection (Figure 4, Appendix A). Moreover, those genes were also regulated by *Bgt* in G24 vs. G0 groups and showed the same expression pattern with H24 vs. H0 groups (Appendix A). The regulation of those genes in response to *Bgt* infection might contribute to the accumulation of G3P and several fatty acids. Moreover, it was demonstrated in our previous study that the levels of G3P, OA18:1, linoleic acid (LA18:2), and α-linolenic acid (ALA18:3) are induced by *Bgt* infection in wheat leaves [23].

### 2.6. Expression Patterns in Response to Glycerol Application (G0 vs. H0)

A total of 414 DEGs (376 up, 38 down) were identified in G0 vs. H0 groups in response to glycerol. Among them, 77 upregulated DEGs and seven downregulated DEGs were overlapped with those identified in H24 vs. H0 and G24 vs. G0 groups (Figure 1D, Appendix A). Those 414 DEGs were annotated in some GO terms such as cell killing, immune system process, and response to stimulus (Appendix A). GO enrichment analysis revealed some important GO terms, including response to JA, response to SA, defense response to bacterium, wounding, lipid oxidation, growth, lateral root formation, oxylipin metabolic process, linoleate 9S-lipoxygenase activity, allene oxide synthase activity, linoleate 13S-lipoxygenase activity, and hydroperoxide dehydratase activity (Figure 5A). KEGG pathway analysis revealed that 21.1% (19/90) of genes in the linoleic acid metabolism, 2.3% (3/131) genes in cutin, suberine, and wax biosynthesis, and 1.9% (5/265) genes in glycerolipid metabolism could be regulated by glycerol (Appendix A). KEGGenrichment analysis showed that the most highly enriched pathways were linoleic acid metabolism and alpha-linolenic acid metabolism (Figure 5B).

The following DEGs induced by glycerol were found in the linoleic acid metabolism pathway: 16 DEGs identified as linoleate 9S-lipoxygenase (*LOX1_5*, 1.13.11.58), and three DEGs identified as lipoxygenase (*LOX2S*, 1.13.11.12) (Appendix A, Table 2). In the alpha-linolenic acid metabolism pathway: two DEGs identified as lipoxygenase (*LOX2S*, 1.13.11.12), seven DEGs identified as allene oxide synthase (hydroperoxide dehydratase, *AOS*, 4.2.1.92), and two DEGs identified as 12-oxophytodienoic acid reductase (*OPRs*, 1.3.1.42) were induced by glycerol (Appendix A, Table 2). Among them, the *AOS* family genes were the most significantly-induced genes from 8.5–20.7 folds by glycerol application, and most were also induced by *Bgt* infection. Traes_2DL_FE7B99D58, identified as probable linoleate 9S-lipoxygenase 5, showed higher expression than other homologous gene members, and it was induced by both glycerol and *Bgt* (Table 2).

Glycerol-induced plant hormone signal transduction pathway was indicated by the upregulation of *AUX/IAA*, *PP2C*, and *JAZ*, annotated in the IAA, ABA, and JA pathways (Appendix A). Glycerol also induced plant–pathogen interaction pathways, such as *HSP90* (two DEGs) (Appendix A). In addition, many pathogenesis-related proteins were induced by glycerol, such as genes encoding PR10, PR1, PRMS, disease resistance protein RPM1, chitinase 1, chitinase 8, HSP70, and almost all of those *PR* genes were also induced by *Bgt* (Table 2 and Appendix A). KEGG enrichment analysis showed that down-regulated DEGs by glycerol were enriched in those pathways of photosynthesis and monoterpenoid biosynthesis (Appendix A). 

### 2.7. Expression Patterns and DEGs in G24 vs. H24 Groups

A total of 308 DEGs were upregulated and 155 DEGs were downregulated in the G24 vs. H24 groups. Among them, 120 upregulated DEGs and two downregulated DEGs overlapped with G0 vs. H0 groups (Figure 6A, Appendix A). The most enriched GO terms of those 308 DEGs in G24 vs. H24 groups were similar to G0 vs. H0 groups, such as response to jasmonic acid, oxylipin metabolic process, lipid oxidation, linoleate 9S-lipoxygenase activity, allene oxide synthase activity, linoleate 13S-lipoxygenase activity, and hydroperoxide dehydratase (Figure 6B). The two pathways of linoleic acid metabolism and alpha-linolenic acid metabolism also was identified from the enriched KEGG pathways in the G24 vs. H24 groups (Figure 6C).

In the alpha-linolenic acid metabolism pathway, 1.13.11.12 identified as a lipoxygenase (*LOX2S*), eight DEGs identified as allene oxide synthase (*AOS*, 4.2.1.92), and 5.3.99.6 identified as an allene oxide cyclase (*AOC*) were upregulated in the G24 pool compared with H24 pool. In the linoleic acid metabolism pathway (ko00591), six DEGs identified as linoleate 9S-lipoxygenase (*LOX1_5*, 1.13.11.58), and 1.13.11.12 identified as a lipoxygenase (*LOX2S*), were upregulated in the G24 pool compared with H24 pool (Appendix A).

In the plant hormone signal transduction pathway, *SnPK2*, and *JAZ* (two DEGs), were upregulated in the G24 pool compared with H24 pool (Appendix A). *PR-1* was downregulated in the G24 pool. In the plant–pathogen interaction pathway, *HSP90* (two DEGs) and *FLS2* were upregulated in the G24 pool (Appendix A). Moreover, those genes encoding peroxygenase, chitinase 8, peroxidase, beta-glucosidase, UDP-glycosyltransferase, glucan endo-1,3-beta-glucosidase, and xylanase inhibitor were upregulated in the G24 pool (Appendix A). We found that most of *AOC*, *LOX*, and *PR* genes were also induced in the G0 pool compared with H0 pool (Appendix A). It suggested that those genes were significantly changed by glycerol application (G24 vs. H24; G0 vs. H0) and might take part in the resistance pathways in the G24 group.

### 2.8. Plant Hormones Cross-Talk in Response to Glycerol Application

*AUX/IAA*, *PP2C*, and *JAZ* were induced by glycerol, which was annotated in the IAA, ABA, and JA pathways, indicating activation of plant hormone signal transduction pathways (Appendix A). Moreover, the enriched GO terms in response to jasmonic acid and salicylic acid were identified in the G0 vs. H0 groups (Figure 5A). These results suggest that glycerol was involved in the regulation of SA, JA, IAA, and ABA pathways. Quantification of plant hormones validated glycerol induction of SA and JA levels, and reduced IAA level, while ABA level showed no change (Figure 7).

## 3. Discussion

Glycerol-induced resistance was reported in different plants to different pathogens as a typical non-host resistance, with upregulation of several *PR* genes and ROS accumulation [15,20,21,22]. ETI is the main mechanism for host resistance, with a race-specific resistance involving cell death. However, some overlapping defense signaling pathways or genes have been identified between non-host and host resistance, such as *PR* genes, oxidative burst-associated genes, and cell defense genes [27]. In this study, to better understand the mechanism, the whole transcriptome analysis of wheat plants infected by *Bgt* after glycerol or water treatment was used to reveal extensive alteration in gene expression. Our results showed that glycerol induced some defense-related pathways or genes which associated with ROS accumulation, cell death, and *PR* genes, which might contribute to resistance to *Bgt* in wheat (Figure 8).

We identified genes involved in the accumulation of G3P and OA18:1, including the upregulation of three *TaGLI1* and three *TaSSI2* homologs and downregulation of *TaACT1*, in response to *Bgt* infection (Figure 3 and Figure 4; Appendix A). Previous reports show that the expression of *SSI2* and *GLI1* genes are induced by powdery mildew both in wheat and barley [23]. G3P and OA18:1 are important signal molecules involved in plant resistance. Mutants defective in G3P synthesis are compromised in systemic acquired resistance (SAR), while exogenous G3P induces SAR [9,13]. Overexpressing *GLI1* increases G3P levels and enhances resistance against bacterial disease in *Arabidopsis* [11]. Hence, the regulation of genes in the glycerol metabolism pathway, *TaGLI1* and *TaACT1*, which was followed by *Bgt* infection, might contribute to G3P accumulation and induction of disease resistance (Figure 3). However, numerous studies showed that downregulation of OA18:1 improves plant resistance [14,15,28]. For example, the *ssi2* mutants or RNAi-mediated knockdown of *SSI2* show enhanced resistance to different pathogens, with a lower OA18:1 level. Our previous results showed that OA18:1, LA18:2, and ALA18:3 levels were induced by *Bgt* infection [23]. It was demonstrated that unsaturated fatty acids (OA18:1, LA18:2, and ALA18:3) can inhibit innate immunity responses related to callose deposition in wheat [29]. Plant fatty acids can also be transferred to the pathogenic fungus and are required for colonization by pathogens as a fungal organic carbon source [19]. In this situation, the role of upregulated *TaSSI2*, *FabI*, *FabZ*, *FabF*, and *FabD*, following *Bgt* infection, might not contribute to wheat resistance (Figure 4). In addition, lipids encompassing fatty acids, fatty acid-based polymers, and fatty acid derivatives are part of the membrane structures of cells and tissues. Upon microbe infestation, the membrane is modified and membrane lipid defense-signaling molecules, including free fatty acids, oxylipins, JA, and the potent second messenger phosphatidic acid are released [30]. However, the role of membrane lipid is still unclear for disease resistance in plants, and the impact on membrane lipid after glycerol application also needs further research.

JA is derived from the fatty acid ALA18:3, and the first step is catalyzed by lipoxygenases (LOXs), as the formation of FA hydroperoxides, then formation of the catalyzed reaction by the AOS, AOC, and OPR3 [31,32]. In our analysis of glycerol application, linolenic acid and alpha-linolenic acid (ALA18:3) metabolism were the two most enriched KEGG pathways (Figure 5B). The upregulations of *LOX2S*, *LOX1_5*, *AOS*, and *OPRs* by glycerol application from those two pathways, might contribute to JA accumulation. Further support for these results is provided in the current study by quantification of plant hormones that showed JA accumulation induced by glycerol treatment. However, JA is involved in the defense against necrotrophic pathogens, preventing plant cell death, and inducing defense responses to restrict further pathogen infection [33]. Treatment with JA is shown to protect plants against herbivore attack and reduce the severity of infection by necrotrophic fungi [34,35]; while the application of MeJA does not induce resistance to powdery mildew (biotrophic fungi) in wheat [36]. Furthermore, we found the JASMONATE ZIM DOMAIN (JAZ) in JA signal transduction pathway was induced by glycerol. JAZ is a key target of microbes in their effort to suppress JA-dependent signaling [37]. Previous results suggest that tomato JAZ proteins regulate the progression of cell death during host and non-host interactions with different bacterial strains [38]. So, the upregulation of JAZ by glycerol might contribute to resistance to *Bgt* in wheat.

Furthermore, we found that glycerol also reduced IAA and induced SA levels. IAA induces the expression of expansins, proteins that loosen the cell wall, which is key for plant growth but may also make the plant vulnerable to biotic intruders. Exogenous application of IAA aggravates some pathogenic diseases progression [39,40]. Auxin-resistant *Arabidopsis* by mutation of genes functioning in auxin signaling shows enhanced disease resistance [41]. Previous reports showed that Aux/IAAs repressor proteins confer an improved disease resistance against pathogens, regulate the transcript levels of *PRs*, ROS accumulation, and callose development in the plant defense response, and act as transcriptional repressors of ARF genes to regulate downstream auxin-regulated genes [41,42,43]. Thus, the low level of IAA and upregulated Aux/IAAs in response to glycerol may improve plant wheat resistance to *Bgt*. SA is recognized as a very important SAR signal molecule, hence, the upregulation of SA by glycerol may also contribute to plant resistance to *Bgt*. SA causes global repression of auxin-related genes, including the TIR1 receptor gene, resulting in stabilization of the Aux/IAA repressor proteins and inhibition of auxin responses; the inhibitory effect on auxin signaling may be a part of the SA-mediated disease resistance mechanism [41]. Moreover, the amount of ABA was not changed after glycerol application, and other plant hormones-related pathways were not significantly changed with transcriptome analysis. Altogether, our analysis indicated that glycerol application altered plant hormone biosynthesis, signaling, and levels involved in *Bgt* resistance mechanisms.

Some pathogenesis related genes were induced by glycerol application such as encoding, PR10, PR1, chitinase-1, chitinase-8, peroxidase, glucan endo-1,3-beta-glucosidase, beta-glucosidase, UDP-glycosyltransferase, pathogenesis-related protein PRMS, disease resistance protein RPM1, heat shock 70 kDa protein, and heat shock 90 kDa protein. Previous reports showed that the downstream genes of SAR induce the expression of some *PR* protein genes, such as *PR1*, *PR2*, chitinase (*PR3*, *PR8*, and *PR11*), peroxidase (*PR9*), and oxalate oxidase (*PR15* and *PR16*), and those *PR* genes can be used as potential candidate genes for improvement of the pathogen resistance of wheat and barley [44]. For example, *PR* genes encoding hydrolytic enzymes chitinases and β-1,3-glucanases, are very important in plants for invading pathogens, and overexpression of some *PRs* genes improves the resistance to different pathogens in different plants [45,46]. These results suggest that up-expression of those *PRs* by glycerol application might contribute to wheat resistance, and those *PRs* could be used as candidate genes for improving wheat resistance.

## 4. Material and Methods

### 4.1. Plant and Fungal Materials

The *Bgt* susceptible bread wheat cultivar Xuezao was used in this experiment. Powdery mildew isolate E09 was provided by Xiayu Duan, Institute of Plant Protection, Chinese Academy of Agricultural Sciences.

### 4.2. Plants Growth and Glycerol Treatments

Plants potted and maintained in a growth chamber with humidity of 75%, 26/20 °C day/night temperature regime, 12:12 h light/dark cycle, and light intensity of 40 µmol m^−2^ s^−1^. Wheat leaves at seedling stage (about one-week-old) were sprayed with 3% glycerol (containing 0.02% Silwett L-77) when the first leaves were fully expanded. Control plants were sprayed with water, containing 0.02% Silwett L-77. Glycerol and water treatments were preformed two times (once a day) until liquid was dropping from the leaves, to make sure sufficient treatments. Each treatment included three pots of plants, and each pot (12 × 12 cm) included 16 plants.

### 4.3. Pathogen Maintenance and Inoculation

Isolate E09 was maintained on the susceptible wheat line Xuezao through weekly transfer to new plants in the growth chamber with the same controls as described in Section 4.2. The fresh conidia were collected on the dark paper and papered before inoculation. After one day of glycerol or water treatments, wheat plants were put into a vaccination tower; conidia were slowly blown into the vaccination tower by using a blower with an estimated density of 100–150 conidia/mm^2^.

### 4.4. Coomassie Blue Staining

For microscopic observations of fungal development, *Bgt*-infected leaf segments were collected at 24 h post powdery mildew infection (hpi) for Coomassie blue staining as described previously [23]. Wheat leaf segments (5 cm in length) were stored in 50% glycerol and examined under an Olympus BX-43 microscope (Olympus Corporation, Tokyo, Japan). Three plants in each treatment and three leaves per plant were used for Comassie staining and the germination/penetration rates were calculated. The germination/penetration rates of conidiospores (number of germinated spores or penetrating spores relative to the total number of spores) were calculated and presented as the means of three independent replicated experiments. Statistical significance was determined by paired Student’s *t* test.

### 4.5. RNA Extraction, cDNA Library Construction, RNA-Seq, and Data Analysis

Four groups of wheat plants were prepared for RNA-seq: H0 and G0 groups represent water or glycerol-treated plants without *Bgt* infection; H24 and G24 groups represent water or glycerol-treated plants infected with *Bgt* one day after glycerol or water treatments. Each group included three biological repetitions. Ten leaves were pooled together at 24 hpi as one biological repetition for RNA extraction. The leaf samples of H0 and G0 groups were also collected at the same time point as controls of H24 and G24 groups at 24 hpi. The total RNA was extracted using RNA pure Plant Kit (Tiangen Biotech, Beijing, China). Sequencing libraries were generated using NEB Next UltraTM RNA Library Prep Kit for Illumina (NEB, Beijing, China) following manufacturer’s recommendations. The parried-end reads were generated on Illumina Hiseq 2500 platform.

Sequencing data were analyzed by using BMKCloud. Trimmomatic was used to remove the adaptors. TopHat2 tools software were used to map and assemble the reads to reference genome IWGSC RefSeq v1.0, [47]. Only reads with a perfect match or one mismatch were further analyzed and annotated based on the reference genome. Gene function was annotated based on the following databases: Nr (NCBI non-redundant protein sequences); Nt (NCBI non-redundant nucleotide sequences); Pfam (protein family); KOG/COG (clusters of orthologous groups of proteins); Swiss-Prot (a manually annotated and reviewed protein sequence database); KO (KEGG ortholog database); GO (Gene Ontology). Differential expression analysis of four groups was performed using the DESeq R package (1.10.1). Genes with *p*-value ≤ 0.05 were assigned as differentially expressed, with |log2 fold change| ≥1.8. GO enrichment analysis of the differentially expressed genes (DEGs) was implemented by the GOseq R packages based on the Wallenius non-central hyper-geometric distribution, which can adjust for gene length bias in DEGs [48]. KOBAS software was used to test the statistical enrichment of differential expression genes in KEGG pathways [49].

### 4.6. Measurements of Endogenous Plant Hormones

The leaf tissues of both the water treatment (control) and glycerol treatment were collected at the seeding stages one day after glycerol and water treatments. Plant hormones (ABA, IAA, JA, and SA) were extracted and measured from leaf tissue (approximately 50 mg of fresh weight) by simplified icELISA method as described previously [50], slightly modified by using a different antibody which was developed in the Wang Baomin’s lab, China Agricultural University.

## 5. Concluding Remarks

*Bgt* infection might contribute to G3P and OA18:1 accumulation in wheat. Glycerol application could improve G3P levels and reduced OA18:1 levels. Furthermore, glycerol altered plant hormones signaling pathways, which were confirmed by increase in JA and SA and reduction of IAA levels. Moreover, glycerol induced the expression of many *PR* proteins. These glycerol-induced changes probably contributed to wheat resistance to *Bgt* (Figure 8).

## Figures and Tables

**Figure 1 ijms-21-00673-f001:**
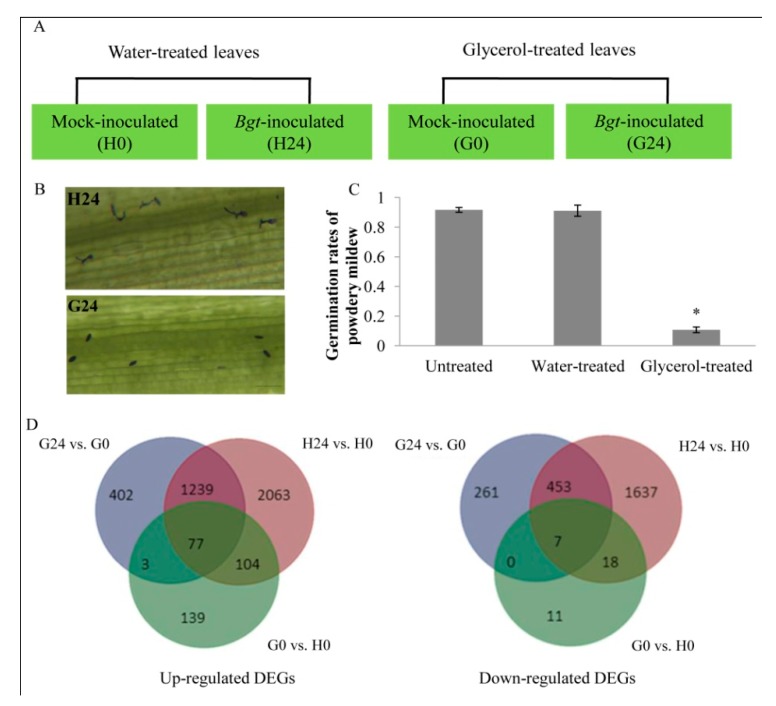
Application of glycerol enhanced *Bgt* resistance in wheat and transcriptome analysis. (**A**) Experimental design. (**B**) Morphology of conidiospore and hyphae in untreated, water-treated, and glycerol-treated Xuezao leaves at 24 h post powdery mildew infection (hpi). (**C**) Germination rates of powdery mildew on the water-treated and glycerol-treated wheat leaves at 24 hpi. Each value is the mean ± standard error (SE) of three independent biological repetitions. Asterisk indicates a significant difference from the germination rates of treated leaves at *p* ≤ 0.05 by Student’s *t*-test. (**D**) Differentially expressed genes (DEGs) overlapped in different groups comparison. H0 and G0 groups presented water or glycerol-treated plants without *Bgt* infection; H24 and G24 group presented water or glycerol-treated plants with *Bgt* infection.

**Figure 2 ijms-21-00673-f002:**
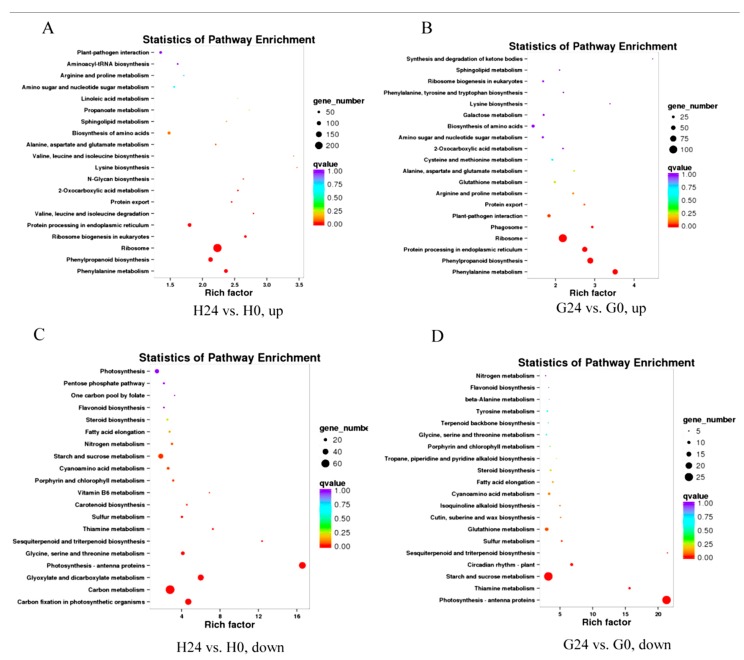
The top 20 enriched KEGG pathways of upregulated (in H24) DEGs in the (**A**) H24 vs. H0 and (**B**) G24 vs. G0 groups. The top 20 enriched KEGG pathways of downregulated DEGs (in H24) in the (**C**) H24 vs. H0 and (**D**) G24 vs. G0 groups. The color represents the Q-value as shown in the legend. Q-values are the *p*-values corrected for multiple hypothesis testing and range from 0 to 1. The closer the Q-value is to zero, the more significant the enrichment. The horizontal axis indicates the rich factor, which means that the ratio of the DEGs number and the number of genes are annotated in this pathway. The greater the rich factor, the greater degree of enrichment. The size of each circle indicates the number of DEGs in that pathway. The larger the circle, the greater number of DEGs.

**Figure 3 ijms-21-00673-f003:**
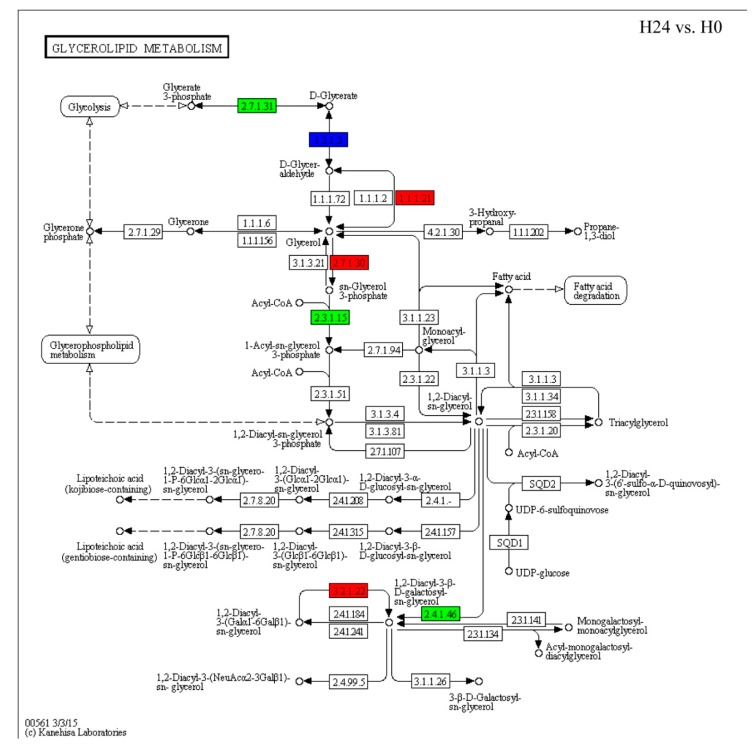
Gene expressions in the glycerolipid metabolism pathway (Ko00561) in response to *Bgt* infection. The red color indicates gene expression induction in H24 group; the green color indicates gene expression reduction in H24 group; the blue color indicates that the expression patterns of some annotated genes were induced and some homologous genes were reduced in H24 group.

**Figure 4 ijms-21-00673-f004:**
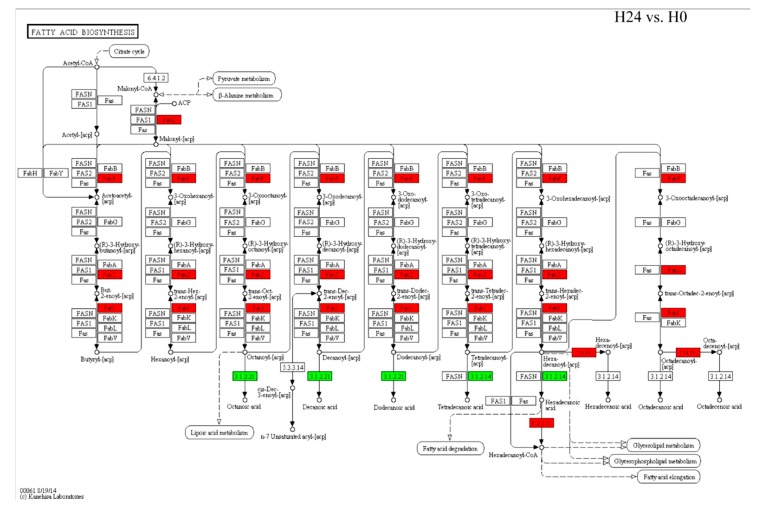
Gene expressions in the fatty acid biosynthesis pathway (Ko00061) in response to *Bgt* infection. The red color indicates gene expression induction by *Bgt*; the green color indicates gene expression reduction by *Bgt*.

**Figure 5 ijms-21-00673-f005:**
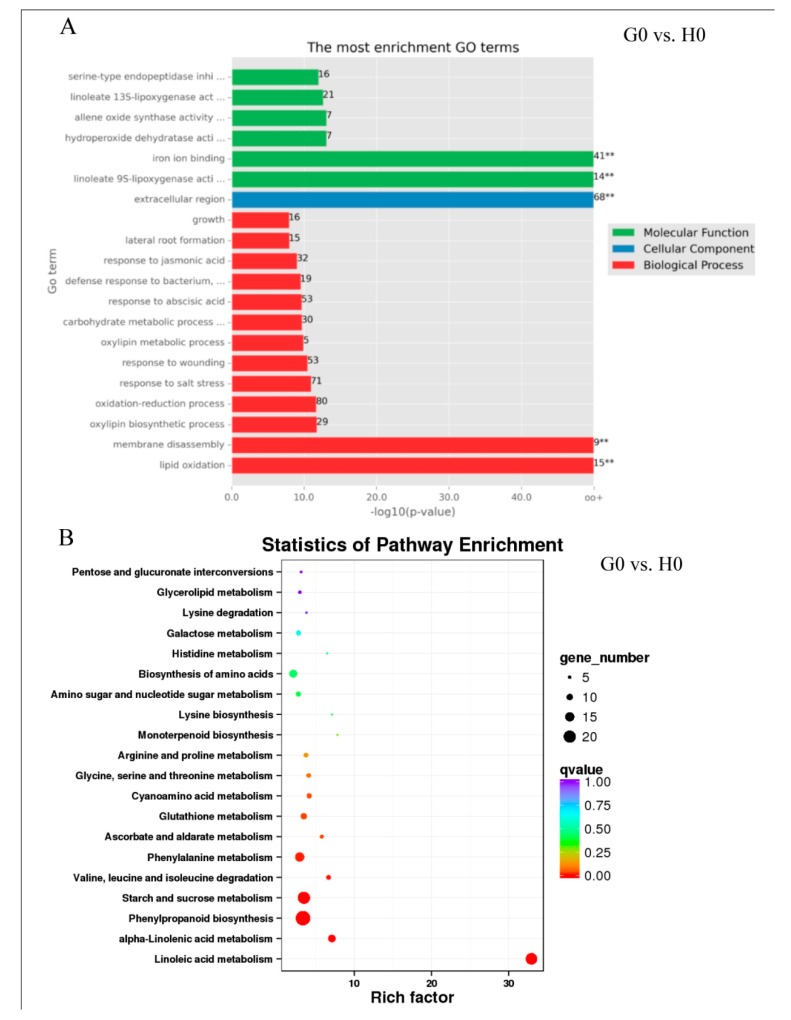
The top 20 enriched GO terms and KEGG pathways. (**A**) GO enrichment analysis of genes differentially expressed in response to glycerol. Data are presented according to the *p*-value. The names of the 20 most highly enriched GO terms are arranged on the vertical axis according to the −log 10 (*p*-value); the horizontal axis represents the −log 10 (*p*-value). ** indicates highly significant differences from H0 group at *p* ≤ 0.01 by Student’s *t*-test. (**B**) Statistical scatterplot showing the pathways enriched in genes differentially regulated in response to glycerol. The color represents the Q-value as shown in the legend. Q-values are *p*-values corrected for multiple hypothesis testing and range from 0 to 1. The closer the Q-value is to zero, the more significant the enrichment. The horizontal axis indicates the rich factor, which means that the ratio of the DEGs number and the number of genes are annotated in this pathway. The greater the rich factor, the greater degree of enrichment. The size of each circle indicates the number of DEGs in that pathway. The bigger the circle, the greater number of DEGs.

**Figure 6 ijms-21-00673-f006:**
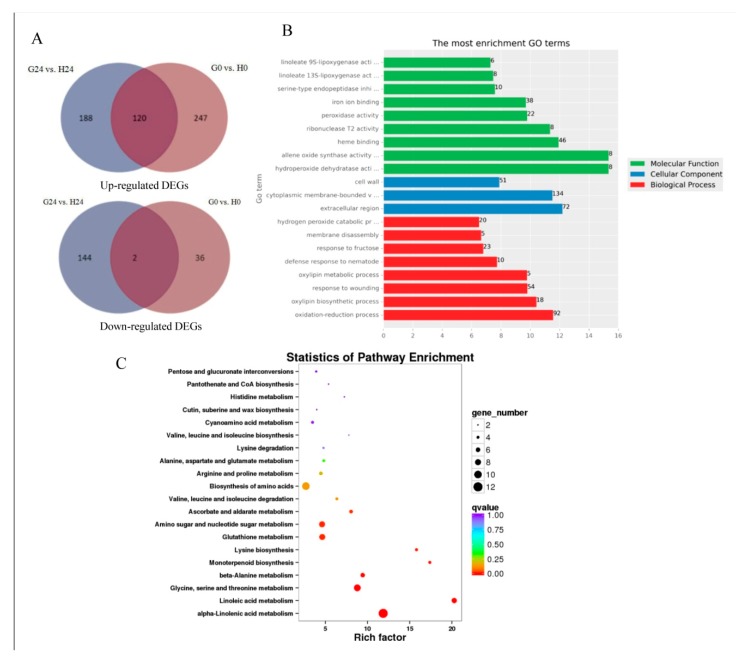
The DEGs in G24 vs. H24 groups. (**A**) DEGs overlapped in different groups comparison. (**B**) GO enrichment analysis of DEGs in G24 vs. H24 groups. (**C**) Statistical scatterplot showing the pathways enriched in genes differentially regulated in response to glycerol in the G24 vs. H24 groups.

**Figure 7 ijms-21-00673-f007:**
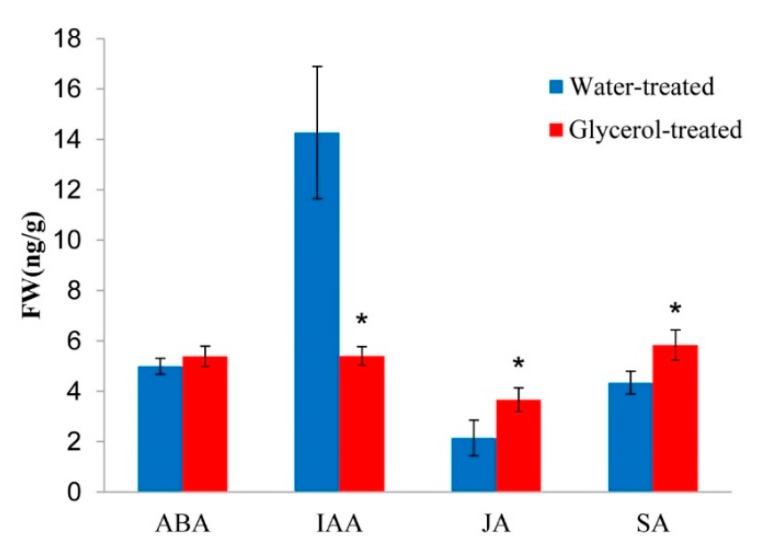
Content of abscisic acid (ABA), auxin (IAA), jasmonic acid (JA), and salicylic acid (SA) levels in water-treated and glycerol-treated Xuezao leaves. Each value is the mean ± SE of three independent biological repetitions. Asterisks indicate a significant difference from the water-treated mock control at *p* ≤ 0.05 determined by Student’s *t*-test.

**Figure 8 ijms-21-00673-f008:**
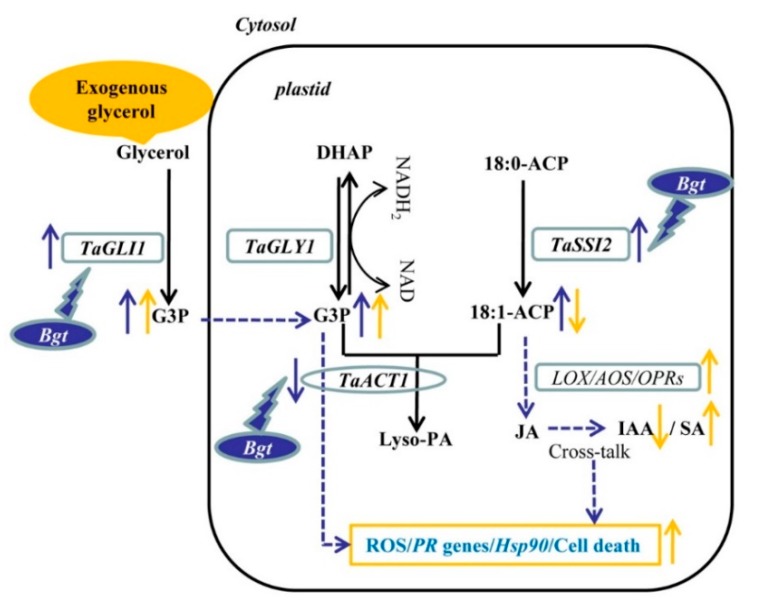
The glycerol-induced resistance pathway. Upon *Bgt* infection, the *TaGLI1* and *TaSSI2* were induced, and *TaACT1* was reduced, which might contribute to G3P and OA18:1 accumulation. G3P accumulation might be an active defense process; OA18:1 accumulation might be not good for plant resistance. Exogenous application of glycerol increased the G3P levels and decreased the level of OA18:1. Glycerol induced some genes involved in JA synthesis (e.g., *LOX*, *AOS*, and *OPRs*), then induced the JA level. Glycerol reduced IAA and induced SA levels with cross-talk of plant hormones. Glycerol induced ROS accumulation, *PR* and *Hsp90* genes expression, and cell death, then contributed resistance to *Bgt* in wheat.

**Table 1 ijms-21-00673-t001:** Comparison of the number of transcripts identified in the DEGs.

DEG Set	All DEGs	Upregulated	Downregulated
H24 vs. H0	5616	3489	2127
G24 vs. G0	2494	1723	771
G0 vs. H0	414	376	38
G24 vs. H24	463	308	155

**Table 2 ijms-21-00673-t002:** DEGs in the G0 vs. H0 groups.

	Gene #ID	H0	G0	H24	G24	Annotation
*1.13.11.12*	Traes_5BS_060785740	**32.93**	**123.12**	46.68	97.83	Lipoxygenase 2.1
	Traes_5DS_E8892706A	**93.92**	**330.19**	155.71	290.37	
	Traes_6DS_7CA5A8F12	**41.62**	**73.37**	30.51	38.32	Lipoxygenase 2.3
*4.2.1.92*	Traes_4AS_41FB87D39	**2.37**	**21.67**	5.56	24.73	Allene oxide synthase 2
	Traes_4AS_9F1B2A7DD	**1.66**	**14.56**	1.00	7.24	
	Traes_4BL_523D155E21	**2.63**	**43.79**	6.79	49.98	
	Traes_4BL_C01E043B6	**1.10**	**10.83**	1.46	7.70	
	Traes_4BL_DD6DD7487	**17.33**	**119.91**	38.96	170.11	
	Traes_4BL_EC6D20026	**5.55**	**115.08**	29.68	122.97	
	Traes_4DL_B3E978E9F	**1.76**	**14.96**	1.02	6.22	
*1.3.1.42*	Traes_6DL_94DCF0B70	**0.29**	**1.18**	2.48	2.35	12-oxophytodienoate reductase
	Traes_7BS_62CC4CA59	**0.63**	**6.32**	0.19	2.01	
*1.13.11.58*	Traes_2DL_17913EE21	**109.35**	**358.45**	275.74	312.34	Linoleate 9S-lipoxygenase 1
	Traes_2DL_CE85DC5C0	**35.10**	**133.18**	93.19	117.98	
	Traes_2DL_D4BCDAA76	**24.12**	**70.55**	41.85	65.25	
	Traes_4BS_63DD9D036	**2.13**	**5.62**	5.75	10.62	
	Traes_4BS_71CB57A0D	**4.33**	**15.90**	8.60	17.88	
	Traes_4BS_939C79184	**1.45**	**9.81**	4.19	9.49	
	Traes_4BS_9DDF3D7C6	**2.05**	**6.42**	5.43	9.57	
	Traes_4DS_7868A8C2E	**2.13**	**11.58**	7.16	18.30	
	Traes_2DL_B5B62EE11	**31.16**	**95.42**	65.15	85.79	Probable linoleate 9S-Lipoxygenase 5
	Traes_2BL_77148B8D8	**18.72**	**76.36**	38.65	61.36	
	Traes_2DL_FE7B99D58	**3700**	**21095**	16544	15919	
	Traes_6AS_9557563D1	**8.17**	**70.75**	6.10	88.52	Putative linoleate 9S-Lipoxygenase 3
	Traes_6BS_B26FD03C8	**7.08**	**62.70**	4.01	41.90	
	Triticum_aestivumLinn_newGene_11732	**1.44**	**13.99**	0.68	17.84	
	Traes_5BL_304FAFA26	**0.09**	**1.78**	0.15	3.64	Linoleate 9S-lipoxygenase 2
	Traes_2AL_5BAB26827	**27.39**	**75.74**	78.10	76.00	Seed linoleate 9S-lipoxygenase-3
*1.11.2.3*	Traes_2AL_6A8D574C4	**0.32**	**3.42**	0.25	2.85	Peroxygenase
	Traes_2BL_5880DAAC3	**0.11**	**1.31**	0.00	0.41	
	Traes_2DL_00618F315	**0.09**	**1.91**	0.10	0.83	
*AUX/IAA*	Triticum_aestivumLinn_newGene_126	**0.70**	**1.47**	0.87	1.62	Auxin-responsive protein IAA18
*PP2C*	Traes_4BL_3403452C0	**0.61**	**2.36**	0.30	0.88	Serine/threonine protein phosphatase 2C 30
*JAZ*	Triticum_aestivumLinn_newGene_18579	**0.50**	**1.32**	0.52	0.92	Protein TIFY 10B
*HSP*	Traes_2AS_67EFE0FAE	**0.24**	**1.53**	0.29	0.79	Heat shock protein 90
	Traes_2DS_3B16D8173	**0.09**	**1.58**	0.42	0.83	
	Traes_1AL_51CED3DBF	**4.02**	**7.93**	5.77	8.92	Heat shock cognate 70 kDa protein 4
	Traes_1AS_5DE9A16CD	**0.86**	**1.90**	2.53	2.77	Heat shock 70 kDa protein 4L
	Traes_3B_B67388A96	**0.33**	**1.89**	0.35	0.87	Heat shock cognate 70 kDa protein
	Traes_3B_FB10B725B	**0.54**	**2.53**	0.46	1.36	Heat shock cognate 70 kDa protein
	Traes_5BL_C318204D2	**1.58**	**3.14**	2.16	3.18	Hsp70 nucleotide exchange factor fes1-like
*PR-1*	Traes_3DL_85AC9E60D	**0.71**	**22.84**	152.38	109.17	Glucan endo-1,3-beta-glucosidase GII
	Traes_3DL_F5930F58D	**2.13**	**36.99**	344.10	210.39	
	Traes_3DL_48F92563F	**5.79**	**18.03**	258.95	136.05	Glucan endo-1,3-beta-glucosidase GIII
	Triticum_aestivumLinn_newGene_5203	**0.92**	**2.32**	35.78	22.19	beta-1,3-glucanase precursor
	Traes_3AL_BCFD5F303	**0.54**	**16.40**	152.58	103.74	beta-1,3-glucanase
	Traes_3B_9F3320C78	**0.12**	**3.59**	38.21	26.41	Glucan endo-1,3-beta-D-glucosidase
	Traes_5DL_ED441B7EB	**1.62**	**3.54**	3.34	5.37	Cell wall beta-glucosidase
	Triticum_aestivumLinn_newGene_1652	**0.13**	**1.50**	35.07	17.75	Glucan endo-1,3-beta-glucosidase 13
	Traes_1BL_A6F7A9A54	**0.60**	**1.39**	0.45	1.03	Glucan endo-1,3-beta-glucosidase 14
*PR-3*	Traes_7AL_FAE816A85	**3.68**	**29.00**	60.31	80.11	Chitinase 1
	Traes_7DL_24AA71860	**1.55**	**11.67**	28.87	35.39	
	Traes_1AL_E96C0662D	**8.97**	**15.92**	297.95	169.09	Chitinase 8
	Traes_5BL_DE0C53CE2	**1.42**	**18.76**	27.60	25.49	
	Traes_5DL_EEF38A7E4	**0.23**	**3.51**	6.41	13.68	
*Callose synthase*	Traes_3DL_9021F9E75	**0.51**	**1.12**	1.11	1.07	Callose synthase 12
	Traes_6BL_2D48C932A	**0.13**	**0.41**	0.42	0.16	Callose synthase 3
	Traes_7DS_B481462CF	**0.56**	**1.19**	1.16	1.06	Callose synthase 10
*Peroxidase*	Traes_2AS_457604359	**0.44**	**4.44**	67.41	50.64	Peroxidase 2
	Traes_2BS_EAB2C09D0	**0.14**	**1.88**	35.62	36.93	
	Traes_2DS_708F03DA3	**20.25**	**50.88**	301.61	183.45	
	Traes_3DL_B38DFDDFF	**0.39**	**11.67**	2.54	22.57	
	Traes_2DS_D76AB139C	**0.25**	**1.08**	141.26	60.39	Peroxidase 1
	Traes_6BS_0BDACE205	**20.88**	**42.56**	197.18	181.62	Peroxidase 6
	Traes_5BL_36EBD512B	**0.20**	**0.94**	0.77	1.91	Peroxidase 35
*Disease-related gene*	Traes_2BL_19C5224BE	**0.41**	**2.35**	0.41	1.24	Pathogenesis-related protein 1
	Traes_2BL_B657F7F3A	**0.00**	**1.43**	0.12	0.77	
	Traes_4AS_C5AE1BBDD	**2.14**	**5.27**	16.46	8.89	
	Traes_4DL_5C688784F	**3.18**	**9.00**	22.61	16.89	
	Traes_6DL_F504536D5	**4.64**	**9.13**	17.02	24.47	Putative disease resistance RGA4
	Traes_7BS_94EB3B3D6	**11.54**	**42.41**	484.99	343.31	Pathogenesis-related protein PRMS
	Triticum_aestivumLinn_newGene_11833	**2.82**	**5.11**	6.73	9.12	Putative disease resistance RPM1
	Triticum_aestivumLinn_newGene_22934	**0.37**	**1.80**	1.28	3.06	Putative LRR receptor-like serine/threonine-protein kinase

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
