# Peer review of "Glycerol-Induced Powdery Mildew Resistance in Wheat by Regulating Plant Fatty Acid Metabolism, Plant Hormones Cross-Talk, and Pathogenesis-Related Genes"

_ijms, 2020, doi:10.3390/ijms21020673_

Round 1

Reviewer 1 Report

The manuscript entitled “Glycerol Induced Powdery Mildew Resistance in Wheat by Regulating Plant Fatty Acid Metabolism, Plant Hormones Cross-talk, and pathogenesis-related genes” describes the changes produced in the gene expression of wheat plants treated with glycerol and inoculated with Blumeria graminis.  The manuscript is interesting and the results and discussion are well presented. However, some aspects could be addressed in order to improve the overall quality of the manuscript. As general comments, authors should extend a little bit the materials and methods section (i.e. the growth condition of the plants are described only in the plants used for maintaining the pathogen). In the same way, the measurement of plant hormones was performed “as described previously [50]”, however, in that article do not measure plant hormones. If the authors adapted the method, please, describe the modifications.

As specific comments:

-authors should revise English grammar and expression in order to avoid small mistakes (such as deference in line 22).

-Please, consider indicating the full name of the pathogen at its first mention in the text.

-On page 7 line 18 and page 10 line 11: authors describe the identification of hydroperoxide dehydratase (AOS, 4.2.1.92). Please consider using the name Allene oxide synthase (AOS).

Author Response

Response to Reviewer 1 Comments

Thanks a lot for your professional Comments, if you need, please see the attached file which is revised version of our manuscript.

As general comments, authors should extend a little bit the materials and methods section (i.e. the growth condition of the plants are described only in the plants used for maintaining the pathogen). In the same way, the measurement of plant hormones was performed “as described previously [50]”, however, in that article do not measure plant hormones. If the authors adapted the method, please, describe the modifications.

Thanks for your comments.

We have added some information of the growth condition of the plants (page 14 line 15-18).

We have added some information of the measurement of plant hormones (page15 line 39-40). But the antibodies were developed in the Wang Baomin’s lab, China Agricultural University, which were used for measurement of plant hormones. The detailed information about the antibody was their secret. Except that all the protocol is same as in the reference paper [50].

4.6. Measurements of endogenous plant hormones

The leaves tissues of both in the water-treatment (control) and glycerol-treatment were collected at the seeding stages one day after glycerol and water treatments. Plant hormones (ABA, IAA, JA, and SA) were extracted and measured from leaf tissue (approximately 50 mg of fresh weight) by simplified icELISA method as described previously [50], with slightly modified by using different antibody which developed in the Wang Baomin’s lab, China Agricultural University.

We have added some information in page 16 line 7 in the part of Acknowledgments. We are grateful to Prof. Wang Baomin for the analysis of plant hormones.

Moreover, we also improved some other parts in material and methods.

As specific comments:

-authors should revise English grammar and expression in order to avoid small mistakes (such as deference in line 22).

Thanks for your comments.

We have changed the “deference” to “defense” (Page 1 line 22; Page 7 line 11).

Some other mistake also have been corrected such as: changed “parents” to “pattern” (in Page6 line 2).

Changed “and” to “which” in Page 2 line 41.

Changed “was” to “were” in Page 3 line 7.

And some other mistakes were corrected in our manuscript and could be found by tracking.

-Please, consider indicating the full name of the pathogen at its first mention in the text.

-On page 7 line 18 and page 10 line 11: authors describe the identification of hydroperoxide dehydratase (AOS, 4.2.1.92). Please consider using the name Allene oxide synthase (AOS).

Thanks for your comments.

We have modified them in page 7 line 22 and page 10 line 11.

Reviewer 2 Report

The manuscript deals with the analysis of the transcriptome changes of wheat following the powdery mildew infection and/or the treatment with glycerol, which was previously proved effective against the disease. The research was conducted scrupulously, though I found that the controls were not properly chosen. In fact, H0 is not the correct control of H24, because the latter was sampled 24 h after H0, therefore there is a difference in time (24) in addition to the difference in Bgt inoculation; the same mistake is for G24 and H24. On the other hand, I believe that this concern did not affect markedly the analysis result. However, an explanation for this choice should be mentioned in the text. The Introduction and Discussion sections are adequately structured and with sufficient information. Maybe, some more links with other transcriptomics studies on wheat/powdery mildew could be reported to show common and diverse results.
English is quite good, but some revision is recommended, including a double-check of typing errors.
In conclusion, I recommend accepting the manuscript with minor revisions.
Other minor comments are reported here below. P1L6: other phytohormones are also involved P2L22: sort the citations P2L33: have been identified P2L40: glycerol and fatty acid metabolism??? P2L41-43: check this sentence for the verb conjugation (in rest of the manuscript as well) P14L18: why were the plants treated twice with water and once with glycerol? P14L24: procedures for inoculum production and inoculation must be described with more details. P14L27: spell out hpi at the first occurrence (check other acronyms as well). Describe why you used Comassie staining. Only the germination of conidia is presented in Fig1C, but not the other parameters reported here. P15L8: how many plants? How many leaves per plant? P15L11: paired-end. ow many reads per sample? P15L12: start a new line here. Describe the procedure used to remove the adaptors. P15L14: why only mìperfect match and one mismatch? Even if you are working with the same cultivar of the reference genome, several mismatches might be expected. Explain your choice. P15L15: describe the procedure for annotation P15L20: DESeq is obsolete. Did you mean DESeq2? P-value or FDR? Why 1.8 log2fc? Fig1: It is worth noting that you did not use the correct controls: H0 is not the correct control of H24, because the latter was sampled 24 h after H0, therefore there is a difference in time (24) in addition to the difference in Bgt inoculation; the same mistake is for G24 and H24!!! Moreover, you decided to split treatment comparisons into Fig1 and Fig6 (Venn Diagrams) for analyzing the DEGs. Also, you performed the gene enrichment analysis using DEGs of each comparison independently, i.e. including DEGs shared among different comparisons. What do you think to emphasize better the DEGs specific (unique) for each comparison? Finally, you could also generate a unique Venn diagram with all comparisons (Fig1+6), and which serves to identify DEGs uniquely related to G24, H24, etc. (even in the supplemental material, if you wish); this would avoid the drag of several shared DEGs in the enrichment analysis. Please comment on this issue. P3L19: you did not do the assembling, but just mapping P3L23: this must be linked to Fig1 P5L13: how many total genes in this pathway? P5L22: parents or pattern? Tab2: 0 is missing in several numbers (e.g. 32.) P12L8: sort the citations P13L14: did glycerol induce other hormones such as ABA or brassinosteroids? In any case, a sentence about this would be useful.

Author Response

Response to Reviewer 2 Comments

Thanks a lot for your professional comments, the attached file is the revised manuscript, if you need to check. 

The research was conducted scrupulously, though I found that the controls were not properly chosen. In fact, H0 is not the correct control of H24, because the latter was sampled 24 h after H0, therefore there is a difference in time (24) in addition to the difference in Bgt inoculation; the same mistake is for G24 and H24. On the other hand, I believe that this concern did not affect markedly the analysis result. However, an explanation for this choice should be mentioned in the text.

Sorry for the unclear information, you misunderstand our method.

If the time is different, I will use that sentence: “before (0) or after (24) Bgt-infection”. Actually, in our manuscript we always say H0 is without or mock Bgt-infection.

The four group samples for RNA-seq were collected at same time point. The leaf samples of H0 and G0 groups were just without Bgt-infection, H24 and G24 groups were inoculated with Bgt.

So the controls are right. This clearer information has been added in the P15L13-L14.

The Introduction and Discussion sections are adequately structured and with sufficient information. Maybe, some more links with other transcriptomics studies on wheat/powdery mildew could be reported to show common and diverse results.

Thanks for your good suggestions.

In the introduction part we have talked some other transcriptomics studies (in the P2L31-36). Actually, we could find some common GO terms and Pathways such as: PR polypeptides, oxidative stress responsive proteins, lipid degradation and metabolic processes, fatty acid catabolic process, and plant hormone signal transduction pathways. This part work also could be found in P12L8-11.

We checked some important genes such as SSI2 and GLI in PLEXdb (Plant Expression Database), and found that the expression of HvSSI2 and HvGLI1 was induced by various powdery mildew isolates in different barley genotypes. We speculate that the role of the SSI2 and GLI1 genes in the powdery mildew-infection might be conserved in some triticeae crops [23].

We added some information in the P12L18-19. Thanks for your good suggestions.

English is quite good, but some revision is recommended, including a double-check of typing errors.

P1L6: other phytohormones are also involved

Sorry, we didn’t want to put so many phytohormones here because we have cited too much reference papers (50 reference papers now).

I have added ABA and with modified reference, “other phytohormones” also added here in P2L6. Thanks for your good suggestions.

P2L22: sort the citations

I modified them as [15, 20-22] inP2L22.

P2L33: have been identified

I have added this sentence “have been identified” in P2L33.

P2L40: glycerol and fatty acid metabolism???

Thanks for your good suggestions. I tried to save words, however, it confused us.

It should be glycerolipid metabolism and fatty acid biosynthesis pathways in P2L40

The same problems were also addressed in P5L11, P5L13 and Table S6.

P2L41-43: check this sentence for the verb conjugation (in rest of the manuscript as well)

I changed “and” to “which” in P2L41.

Changed “and” to “with the case of” in P2L16.

And some other changes could be found by tracking.

P14L18: why were the plants treated twice with water and once with glycerol?

Both of them treated twice times. I have added more clear information in P14L20-21.

Glycerol and water treatments were preformed two times (once a day) until liquid was dropping from the leaves, to make sure sufficient treatments.

P14L24: procedures for inoculum production and inoculation must be described with more details.

Thanks for your good suggestions; we have added more information in P14L24-31.

Isolate E09 was maintained on the susceptible wheat line Xuezao through weekly transfer to new plants in the growth chamber with same controls as described in 4.2. The fresh conidia were collected on the dark paper and papered before inoculation. After one day of glycerol or water treatments, wheat plants were putted into a vaccination tower; conidia were slowly blowing into vaccination tower by using a blower with a estimated density of 100-150 conidia/mm2.  

P14L27: spell out hpi at the first occurrence (check other acronyms as well). Only the germination of conidia is presented in Fig1C, but not the other parameters reported here.

Actually the first hpi was shown in P3L11 in the Figure1 legend “24 hours post powdery mildew-infection (hpi)”. Anyway, we should add it in the main text as the first occurrence in P15L1.

Describe why you used Comassie staining.

The Comassie staining was used for microscopic observations of fungal development. We have added the information in P15L1-2.

P15L8: how many plants? How many leaves per plant?

Three plants in each treatment and three leaves per plant were used for Comassie staining. I have added this information in P15L5-6.

P15L11: paired-end. how many reads per sample?

The information was showed in Table S1.

P15L12: start a new line here. Describe the procedure used to remove the adaptors.

OK I start a new line here in P15L20.

"Trimmomatic" (http://www.usadellab.org/cms/?page=trimmomatic) were used for remove the adaptors. We have added the information in P15L21-22.

P15L14: why only mìperfect match and one mismatch? Even if you are working with the same cultivar of the reference genome, several mismatches might be expected. Explain your choice. P15L15: describe the procedure for annotation.

Because the wheat genome it is very complex and the annotation is not perfect. In total of five databases (Nr; Pfam; KOG/COG; Swiss-Prot; KO; GO) were used for annotation of RNA-seq data.

For the each reads (125 bp), we think one mismatches is OK, that might be some natural SNPs or sequencing mistakes. But the probability should be very low if there were 2 or more than two SNPs in a short target sequence (125bp). So we chose the prefect and one mismatch.

P15L20: DESeq is obsolete. Did you mean DESeq2? P-value or FDR? Why 1.8 log2fc?

Actually we did sequence and analysis work in 2016, so it is a litter earlier. That is why we used DESeq which is obsolete. But we believe that our result still could be used for analysis. In case lost some important genes, so we chose log1.8fc to get more DEGs.

Fig1: It is worth noting that you did not use the correct controls: H0 is not the correct control of H24, because the latter was sampled 24 h after H0, therefore there is a difference in time (24) in addition to the difference in Bgt inoculation; the same mistake is for G24 and H24!!!

Sorry for the unclear information, you misunderstand our method.

Actually, the leaf samples of H0 and G0 groups were also collected as the same time point as controls of H24 and G24 at 24 hpi. So our controls are no problem.

Moreover, you decided to split treatment comparisons into Fig1 and Fig6 (Venn Diagrams) for analyzing the DEGs.

Because we used four groups of RNA-seq data to identified DEGs, and different treatments have different controls. We want show more clear results, so we separated them.

Firstly, we want to analysis the genes and pathways which were in response to Bgt infection (H24 vs. H0 and G24 vs. G0). Secondly, we found that almost half of genes which were regulated by glycerol (G0 vs H0) were overlapped with those DEGs which were in response to Bgt infection. So we think glycerol could regulate some defense-response genes, that is the story in the F1D. In additionally, we want analyzed the expression patterns and DEGs in G24 vs. H24 group, so we make a Venn Diagram (G24 vs. H24 compared with G0 vs H0 group), to show some overlapped genes which were response to glycerol both with or without Bgt infection. That is the story of Fig 6A.

Also, you performed the gene enrichment analysis using DEGs of each comparison independently, i.e. including DEGs shared among different comparisons. What do you think to emphasize better the DEGs specific (unique) for each comparison?

Thanks for your good suggestions.

If I was not wrong, I think you were meaning two result parts, one is expression patterns in response to Bgt infection (H24 vs. H0 and G24 vs. G0, Fig2) and another result part expression patterns and DEGs in G24 vs. H24 group (Fig 6A). We were also thinking of that to do some enrichment analysis for some DEGs specific (unique), i.e. those 402 DEGs in Fig 1D and 188 DEGs in Fig 6A. We used website App to analysis our seducing data (added in P15L21), now the company close our service after one year used time. So it is very hard for us to add this analysis of DEGs specific (unique) for each comparison now. But from the Fig2, we still can identify some different specific enriched GO terms by compared with different groups (such as shown in P4L20). However, it is very hard for me to understand the meaning of 188 DEGs in Fig 6A (for example), because I have not enough confidence to say those genes is very important for induced resistance or just induced by glycerol with or without Bgt-infection… .Moreover, if readers want focus on specific genes or pathways, they can found some more information form Tables, we hope that could be helped.

Finally, you could also generate a unique Venn diagram with all comparisons (Fig1+6), and which serves to identify DEGs uniquely related to G24, H24, etc. (even in the supplemental material, if you wish); this would avoid the drag of several shared DEGs in the enrichment analysis. Please comment on this issue.

Thanks for your good suggestions.

The same reasons were mentioned in the last two answers (sentence in blue color). 

P3L19: you did not do the assembling, but just mapping

We did assembling by using the wheat reference genome (IWGSC RefSeq v1.0, http://www.wheatgenome.org/). We have added the information in P15L24, actually, we also showed the information in P3L21 (Following assembly,…)

P3L23: this must be linked to Fig1

Thanks for your suggestions.

We have added the Fig1 linked in P3L24.

P5L13: how many total genes in this pathway?

13 DEGs from 265 genes which were involved in glycerolipid metabolism pathways were regulated by Bgt infection (Figure 3, Table S5, S6).

Moreover, 11 DEGs from 265 genes which were involved in fatty acid biosynthesis pathways were regulated by Bgt infection ( Figure 3, Table S5, S6).

We have added this information in P5L15 and P521.

Actually, that information could be found in Table S5. But it is better to add it in the main text.

P5L22: parents or pattern?

Thanks for your suggestions.

We have corrected it. It should be pattern (in P6L2).

Tab2: 0 is missing in several numbers (e.g. 32.)

I am sorry, I don’t understand that comment. I think maybe you mean the numbers in Table? e.g. 32. Should be 32.0?. I think the table has been changed by editor. It is OK, or the editor will order it later.

P12L8: sort the citations

I have sorted the citations as [15, 20-22] in P12L8.

P13L14: did glycerol induce other hormones such as ABA or brassinosteroids? In any case, a sentence about this would be useful.

Thanks a lot for your suggestions.

The glycerol did not change the amount of ABA, which show in Fig 7. We haven’t checked the amount of brassinosteroids.

Furthermore, from the enrichment analysis of GO terms and DEGG pathways, we did not identify some DEGs which related other plant hormones.

We have added this information in the P13L30-31.